# Common Bean Yield and Zinc Use Efficiency in Association with Diazotrophic Bacteria Co-Inoculations

Arshad Jalal [1], Fernando Shintate Galindo [2], Eduardo Henrique Marcandalli Boleta [1], Carlos Eduardo da Silva Oliveira [1], André Rodrigues dos Reis [3], Thiago Assis Rodrigues Nogueira [1], Mário João Moretti Neto [1], Emariane Satin Mortinho [1], Guilherme Carlos Fernandes [1] and Marcelo Carvalho Minhoto Teixeira Filho [1,*]

[1] Department of Plant Protection, Rural Engineering and Soils (DEFERS), São Paulo State University (UNESP), Ilha Solteira 15385-000, SP, Brazil; arshad.jalal@unesp.br (A.J.); eduardomarcandalli7@gmail.com (E.H.M.B.); ces.oliveira@unesp.br (C.E.d.S.O.); tar.nogueira@unesp.br (T.A.R.N.); mario.moretti@unesp.br (M.J.M.N.); emariane.satin@unesp.br (E.S.M.); guilherme.carlos.fernandes@gmail.com (G.C.F.)
[2] Center for Nuclear Energy in Agriculture (CENA), University of São Paulo (USP), Piracicaba 13416-000, SP, Brazil; fs.galindo@yahoo.com.br
[3] School of Sciences and Engineering, São Paulo State University (UNESP), Tupã 17602-496, SP, Brazil; andre.reis@unesp.br
* Correspondence: mcm.teixeira-filho@unesp.br

**Abstract:** Enrichment of staple food with zinc (Zn) along with solubilizing bacteria is a sustainable and practical approach to overcome Zn malnutrition in human beings by improving plant nutrition, nutrient use efficiency, and productivity. Common bean (*Phaseolus vulgaris* L.) is one of a staple food of global population and has a prospective role in agronomic Zn biofortification. In this context, we evaluated the effect of diazotrophic bacterial co-inoculations (No inoculation, *Rhizobium tropici*, *R. tropici* + *Azospirillum brasilense*, *R. tropici* + *Bacillus subtilis*, *R. tropici* + *Pseudomonas fluorescens*, *R. tropici* + *A. brasilense* + *B. subtilis*, and *R. tropici* + *A. brasilense* + *P. fluorescens*) in association with soil Zn application (without and with 8 kg Zn ha$^{-1}$) on Zn nutrition, growth, yield, and Zn use efficiencies in common bean in the 2019 and 2020 crop seasons. Soil Zn application in combination with *R. tropici* + *B. subtilis* improved Zn accumulation in shoot and grains with greater shoot dry matter, grain yield, and estimated Zn intake. Zinc use efficiency, recovery, and utilization were also increased with co-inoculation of *R. tropici* + *B. subtilis*, whereas agro-physiological efficiency was increased with triple co-inoculation of *R. tropici* + *A. brasilense* + *P. fluorescens*. Therefore, co-inoculation of *R. tropici* + *B. subtilis* in association with Zn application is recommended for biofortification and higher Zn use efficiencies in common bean in the tropical savannah of Brazil.

**Keywords:** *Phaseolus vulgaris* L.; diazotrophic bacteria; zinc fertilization; Zn uptake; Zn biofortification

## 1. Introduction

Common bean (*Phaseolus vulgaris* L.) originated in central Mexico and is now grown globally, especially in Central and South America [1]. Beans have imperative traditional, historical, and nutritional profiles (proteins, amino acid, and minerals, as well as antioxidants and polyphenols) that fulfill the nutritional needs of human diet in many regions of the world as a staple food source [2,3]. The global annual average bean production is about 26.5 million tons [4]. The mean production of beans in Brazil has increased by 3.15 million tons in the preceding 20 years, being adjusted to Brazil demand [5]. Legumes improve soil fertility by fixing nitrogen (N), enhancing microbial activity, and decreasing dependency on mineral fertilizers, which helps to promote a sustainable environment and production [6].

Zinc (Zn) is one of the most important micronutrients for all living organisms [7], including plants, humans, and micro-flora [8] and is required throughout their life cycles in small quantities to orchestrate a complete array of physiological functions [9]. Zinc deficiency has frequently been reported in tropical crops [10], concurrently being declared as

"hunger of the day", with several health issues [11], especially in developing countries [12]. Zinc deficiency affects almost 17.3% of the global population and 30% of South Asian countries [13], and therefore affects 2 billion people within the global population, and it is ranked as the fifth highest health risk factor in developing countries [14,15]. The inadequate dietary status of Zn is anti-proportional to human health and leads to several diseases [16], including immune deficiency syndrome, pneumonia, memory disorder, cancer, respiratory and cardiovascular disorder, and diarrhea in humans [17,18]. In the current scenario, it is interesting that Zn deficiency may be one of the predisposing factors for the infection and progression of COVID-19 [19].

Zinc deficiency in crop plants is an alarming concern and is the most effective micronutrient limiter in legume yields. Zinc plays a critical role in several plant physiological processes, including protein synthesis [20], energy production, maintenance of membrane integrity, and cell growth and multiplication [21]. Zinc also has a role in different photosynthetic and enzymatic activities such as peptidases, dehydrogenases, phosphohydrolases, and pollen fertility [22,23]. It is therefore important to combat Zn deficiency in soil, plants, and most importantly, in human beings [24,25]. Soil Zn application is the most practiced strategy in field crops to improve yield and grain Zn concentration [26]. The deficiency of Zn cannot be treated solely with Zn application for better crop establishment, productivity, and Zn use efficiency, especially in tropical regions, due to complexation with carbonates and oxides [27]. Therefore, new alternatives and sustainable strategies need to be adapted for better nutrient replenishment and productivity with higher Zn use efficiency (ZnUE).

Nutrients enriched plant rhizosphere environments significantly stimulate several ecological processes such as decomposition of organic matter, homeostasis, and nutrient cycling to reduce crop dependency on synthetic fertilizers to support sustainable and stable ecosystems [28]. Plant growth-promoting bacteria (PGPBs) or diazotrophic bacteria adapt several direct and indirect mechanisms to improve plant growth [29]. They improve nutrient availability by playing a role in biological nitrogen fixation (BNF), nutrient solubilization, and enzymes synthesis through direct mechanisms [29–31] while inhibiting pathogen infestation by producing siderophores and antibiotics [32]. Zinc solubilizing bacteria could be applied in one or more of the above mentioned mechanisms to increase Zn solubility through production of organic and inorganic acids and several chelators [33]. A diverse range of bacteria including species of *Rhizobium*, *Pseudomonas*, *Azospirillum*, *Azotobacter*, *Bacillus*, *Enterobacter*, *Acinetobacter*, and many others may solubilize or tolerate Zn and plant growth promoters [23,34,35].

The effects Zn biofortification of the interaction of diazotrophic bacteria in co-inoculation with soil applied Zn is lacking in the literature. There is a research gap on the association of diazotrophic bacterial co-inoculation and Zn fertilization on Zn nutrition, ZnUE, and yields of common bean. It is necessary to determine such strategies to improve the Zn nutritional quality of common bean to increase the dietary intake of the population. Therefore, the hypothesis of this study was that there may be a synergetic association of different diazotrophic bacteria with soil Zn application on Zn availability on plant and soil, ZnUE, yield, and intake of fortified common bean in the tropical savannah of Brazil. The objectives of the study were to evaluate the effect of soil Zn application in combination with seed inoculation of different diazotrophic bacteria on common bean growth and yield. Additionally, the effect of co-inoculations and Zn fertilization on Zn accumulation in plant shoot and grains, Zn intake, and Zn use efficiencies for sustainable biofortification in the tropical savannah of Brazil.

## 2. Materials and Methods

### 2.1. Experimental Area and Location

A field experiment on common bean was performed during two consecutive cropping years (2019 and 2020) at the research farm of the School of Engineering (UNESP) at the geographical coordinates 20°22′ S, 51°22′ W, and 335 m altitude (Figure 1), located in Mato Grosso do Sul, Brazil. The soil is classified as Rhodic Haplustox with clay texture [36].

The experimental site has been used for the cultivation of an annual crop (cereals and legumes) for more than 28 years, the last 12 being under no tillage [37]. The climate of the experimental area is classified as Aw according to Köppen classification, and the data of both cropping seasons are summarized in Figure 2.

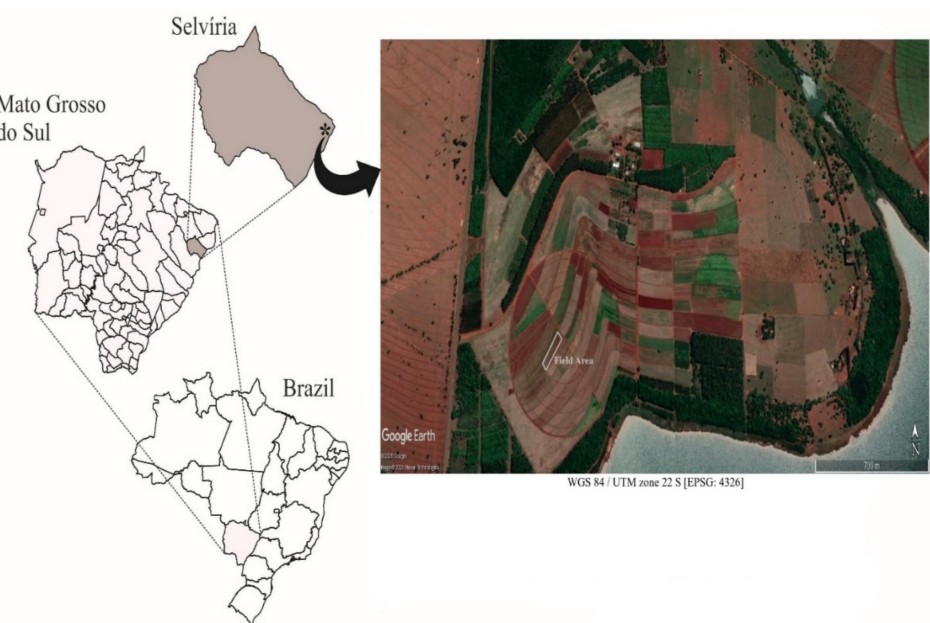

**Figure 1.** Location of the experimental area at the Research and Extension Farm, UNESP—Ilha Solteira Campus, at Selvíria—Mato Grosso do Sul state, Brazil (20°22′ S, 51°22′ W, altitude of 335 m) in 2019 and 2020 crop seasons. The map was created by using geographic information system (QGIS) software and the Google Earth program. The QGIS Development Team (2021). Open Source Geospatial Foundation project. http://qgis.osgeo.org. Accessed on: 9 March 2021. Projection System WGS 84/UTM 200DC [EPSG: 4326]. This image was taken from the Google Earth program, Google Company (2021). Map data: Google, Maxar Technologies.

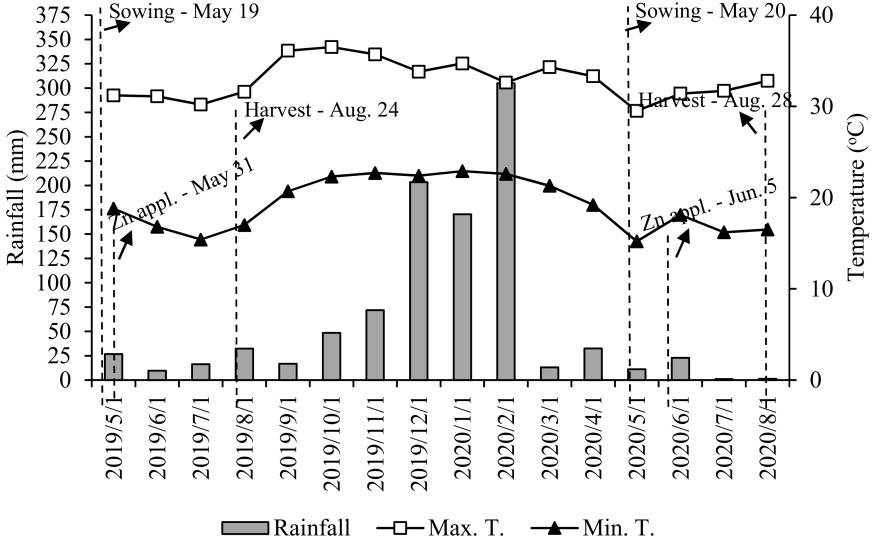

**Figure 2.** Rainfall maximum, average, and minimum temperatures and air relative humidity were acquired from the weather station of the Education and Research Farm of the Faculty of Engineering—UNESP during the common bean cultivation period from May to August 2019 and May to August 2020.

### 2.2. Soil Analysis

A composite soil sample was collected from the experimental site in the soil layer of 0–0.20 m depth for the pre-experimental soil physio-chemical analysis, following the standard procedures of Raij et al. [38]. Table 1 indicates the physio-chemical attributes of the study area, which is clayey and slightly acidic with a minimal Zn concentration.

**Table 1.** Analyzed results of soil sample (0–0.20 m) from experimental site before experiment implementation. UNESP—Ilha Solteira, Selvíria, state of Mato Grosso do Sul, Brazil, 2019.

| Properties | Units | Values |
|---|---|---|
| Clay | g kg$^{-1}$ | 433 |
| Sand | g kg$^{-1}$ | 471 |
| Silt | g kg$^{-1}$ | 90 |
| pH (CaCl$_2$) | — | 5.2 |
| Organic matter | mg dm$^{-3}$ | 18 |
| P (resin) | mg dm$^{-3}$ | 38 |
| K | mmol$_c$ dm$^{-3}$ | 1.7 |
| Ca | mmol$_c$ dm$^{-3}$ | 21 |
| Mg | mmol$_c$ dm$^{-3}$ | 15 |
| B (hot water) | mg dm$^{-3}$ | 0.14 |
| Cu (DTPA) | mg dm$^{-3}$ | 3.4 |
| Fe (DTPA) | mg dm$^{-3}$ | 25 |
| Zn (DTPA) | mg dm$^{-3}$ | 0.9 |
| Mn (DTPA) | mg dm$^{-3}$ | 38.1 |
| S-SO$_4$ | mg dm$^{-3}$ | 4.0 |
| H + Al | mmol$_c$ dm$^{-3}$ | 34 |
| CEC (pH 7.0) | mmol$_c$ dm$^{-3}$ | 75.7 |
| V | % | 50 |

CEC: cation exchange capacity; V: base saturation; DTPA: diethylenetriaminepentaacetic acid.

### 2.3. Experimental Design and Treatments

The experimental design for the common bean crop was randomized, with complete blocks having a 7 × 2 factorial scheme with four replications. The experimental factors were comprised of bacterial seeds inoculations (1—No inoculation, 2—*Rhizobium tropici*, 3—*R. tropici + Azospirillum brasilense*, 4—*R. tropici + Bacillus subtilis*, 5—*R. tropici + Pseudomonas fluorescens*, 6—*R. tropici + A. brasilense + B. subtilis*, and 7—*R. tropici + A. brasilense + P. fluorescens*), without and with soil applied Zn (0 and 8 kg Zn ha$^{-1}$).

The inoculation of common beans with *R. tropici* was carried out using a commercial peat inoculant, strain SEMIA 4080, with $2 \times 10^9$ colony forming units (CFU) g$^{-1}$ at a dose of 200 g for every 100 kg of seeds. To facilitate inoculant adhesion with the seeds, a 10% sugar solution was used to properly and homogeneously mix the seeds. The product is commercially registered with the Ministry of Agriculture and Livestock, Brazil. The inoculation of bacterium *A. brasilense* strains Ab-V5 and Ab-V6 (strains CNPSo 2083 = Ab-V5 and CNPSo 2084 = Ab-V6 with a guarantee of $2 \times 10^8$ CFU mL$^{-1}$) was performed at a dose of 300 mL of inoculant (liquid) per hectare of sown seeds. The inoculation of *B. subtilis* (strain CCTB04 with a guarantee of $1 \times 10^8$ CFU mL$^{-1}$) and *P. fluorescens* (strain CCTB03 with a guarantee of $2 \times 10^8$ CFU mL$^{-1}$) were applied at a dose of 150 mL ha$^{-1}$, according to the recommendation of the inoculant providing company@Total Biotechnology, Curitiba, Brazil. Inoculation was performed an hour before plantation of the crop.

Zinc was applied from a source of zinc sulfate (21% Zn and 11% of S) to the soil surface by side dressing cover. Zinc doses (0 and 8 kg ha$^{-1}$) were manually applied with an even distribution and without incorporation on the soil surface. The Zn dose applied to the soil is based on [39,40], who recommended 5 to 30 kg ha$^{-1}$ to the soil. The amount of Zn per treatment per plot was applied at the V1/V2 stage (1/2 trifoliate leaves completely unfolded) in both 2019 and 2020 cropping seasons. The experimental area was irrigated

with a central pivot irrigation system (14 mm) to evenly incorporate the Zn fertilizer into the soil.

### 2.4. Plant Materials

The sowing bed was sprayed with systemic (selective) herbicide (2, 4-D—670 g ha$^{-1}$ of the active ingredient (AI)) and a broad-spectrum herbicide (glyphosate, 1800 g ha$^{-1}$ of AI) 15 days before the experiment implementation for controlling already emerged narrow and broad leaved weeds. The common bean cultivar (IPR—Campos Gerais) with potential quality and production was sown in May, 2019, and repeated in May, 2020, on no-tillage beds using the drill sowing method. The seeds of beans were treated with piraclostrobin (2.5% m v$^{-1}$), thiophanate methyl (22.5% m v$^{-1}$), and fipronil (71.3% m v$^{-1}$), based on the recommended dose of the cultivar. The plots were composed of six lines with a plot size of 2.7 × 4.5 m, totalizing 12.15 m$^2$. The recommended dose of NPK was applied at the time of sowing, with 40 kg N ha$^{-1}$ from urea source, 80 kg P$_2$O$_5$ ha$^{-1}$ from triple superphosphate source, and 40 kg K$_2$O ha$^{-1}$ as potassium chloride. The recommended dose of 90 kg N ha$^{-1}$ was applied from ammonium sulphate fertilizer after 30 days of emergence. The experimental area was limited with B, according to the interpretation of the Campinas Agronomic Institute (IAC) [38], so the total experimental area was treated with 1 kg B ha$^{-1}$ as boric acid through a tractor sprayer machine. Irrigation by sprinkling was done with a central-pivot irrigation system, according to the crop need.

### 2.5. Evaluations and Analysis

#### 2.5.1. Zinc Soil and Plant Nutritional Analysis

The Zn soil analysis was performed three days after harvest in both the 2019 and 2020 study years. Five samples were collected with an auger (0.0–0.20 m) from each treatment of each replication and mixed together to obtain a uniform sample of each treatment. The homogeneous sample was collected in separate and already labeled bags. The collected samples were air dried, sieved with a sieving net of 2 mm, and stored at room temperature until the Zn analysis. The Zn analyses were performed as described by Raij et al. [38].

The plant material (leaf, straw, and grain) was collected in proper labeled paper bags and dried in an air-tight oven at 60 ± 5 °C for 72 h until it attained a uniform humidity. The material was then ground in a stainless-steel Wiley knife mill by passing it through a 10-mesh sieve and was then placed in labeled plastic containers. Each sample was weighed (0.25 g), digested with nitroperchloric digestion (HNO$_3$:HClO$_4$ solution), and quantified by atomic absorption spectrophotometry. The analysis was developed following the methodology of Malavolta et al. [41].

#### 2.5.2. Shoot Dry Matter and Yield

Plant height at maturity was determined with a ruler from the ground to the upper apex. Shoot dry matter was determined after harvest of four useful central lines. Common bean were harvested and packed in jute bags and then dried in the shade for approximately 1 week. Each plot sample was threshed with an electric thresher to attain the weight of the processed grains for calculating yield ha$^{-1}$ (productivity at 13% moisture content). After drying, the beans were ground in a Wiley mill for analysis of nutrients.

#### 2.5.3. Zn Plant Accumulation

The Zn accumulation in shoot and grains (g ha$^{-1}$) was calculated via the following formula:

$$ZnSA = \frac{\text{Zinc concentration in shoot} \times \text{dry matter}}{1000} \tag{1}$$

$$ZnGA = \frac{\text{Zinc concentration in grains} \times \text{grain yield}}{1000} \tag{2}$$

where ZnSA = Shoot Zn accumulation and ZnGA = Grain Zn accumulation.

### 2.5.4. Zinc Partitioning and Intake

Zinc partitioning index (ZPI) toward grains and intake were calculated following standard methodology of [42,43]:

$$ZPI = \frac{Zinc\ concentration\ in\ grains}{Zinc\ concentration\ in\ shoot} \times 100 \tag{3}$$

$$Zn\ intake = [Zn] \times C \tag{4}$$

where Zn intake (g person$^{-1}$ day$^{-1}$) is the daily Zn intake of an estimated person$^{-1}$, [Zn] (g kg$^{-1}$) is the Zn concentration in biofortified grains from the current study results, and C (kg person$^{-1}$ day$^{-1}$) is the mean consumption of common bean grains per person in Brazil.

### 2.5.5. Zinc Use Efficiency

The following Zn use efficiencies were calculated following measurements based on the standards of [44] via the formula:

$$ZnUE = \frac{Grain\ yield\ ZnF - Grain\ yield\ ZnW}{Applied\ Zn\ dose} \tag{5}$$

$$APE = \frac{Grain\ yield\ ZnF - Grain\ yield\ ZnW}{ZnA\ in\ grain\ and\ shoot\ ZnF - ZnA\ in\ grain\ and\ shoot\ ZnW} \tag{6}$$

$$RAZn\ (\%) = \frac{ZnA\ in\ grain\ and\ shoot\ ZnF - ZnA\ in\ grain\ and\ shoot\ ZnW}{Applied\ Zn\ dose} \tag{7}$$

$$UE = PE \times RAZn \tag{8}$$

where ZnUE = Zinc use efficiency, APE = Agro-physiological efficiency, RAZn = Recovery of applied Zn, UE = Utilization efficiency, PE = physiological efficiency, ZnF = Zn fertilized treatments, ZnW = without Zn fertilized treatments, and ZnA = Zn accumulated.

### 2.6. Statistical Analysis

All data were initially tested for normality using Shapiro and Wilk tests, which showed that the data were normally distributed (W ≥ 0.90). Levene's homoscedasticity tests ($p \leq 0.05$) were performed to access the equality of variances. Afterwards, data were subjected to an analysis of variance (F test). The Zn soil application and diazotrophic bacterial inoculations and their interactions were considered fixed effects in the model. When a main effect or interaction was observed to be significant by the F test ($p \leq 0.05$), the Tukey test ($p \leq 0.05$) was used for comparison of the means of Zn soil application, whereas the Scott Knott test ($p \leq 0.05$) was used for comparison of diazotrophic bacterial inoculations using the ExpDes package in R software (R Development Core Team, 2015). The graphics are made in sigma-plot 12.5.

Pearson correlation analysis ($p \leq 0.05$) was performed using R software (R Development Core Team). To create a heatmap, the corrplot package was used, using the "color" and "cor.mtest" functions to calculate the coefficients and *p*-value matrices. Asterisks were added to the heatmap cells to identify significant correlations.

### 3. Results

#### 3.1. Zinc Nutrition in Soil, Plants and Grains

The plant and grain Zn concentration of common bean were improved with the application of Zn to the soil in top dressing and co-inoculations of different diazotrophic bacteria (Table 2).

**Table 2.** Zinc concentration in soil, leaf, shoot, and grain of common bean under the influence of diazotrophic bacteria and soil applied zinc doses. Selvíria—MS, Brazil, 2019 and 2020.

| Treatments | Zn-Soil Concentration | | Zn-Leaf Concentration | | Zn-Shoot Concentration | | Zn-Grains Concentration | |
|---|---|---|---|---|---|---|---|---|
| | mg dm$^{-3}$ | | mg kg$^{-1}$ | | | | | |
| | **2019** | **2020** | **2019** | **2020** | **2019** | **2020** | **2019** | **2020** |
| Zinc (Zn) application (kg ha$^{-1}$) | | | | | | | | |
| 0 | 2.8 b | 4.6 b | 35.9 b | 42.7 b | 30.7 b | 41.1 b | 47.4 b | 54.0 b |
| 8 | 3.7 a | 6.5 a | 38.2 a | 46.4 a | 38.6 a | 46.6 a | 48.6 a | 56.5 a |
| Diazotrophic bacterial inoculations (I) | | | | | | | | |
| Without | 2.7 c | 4.8 c | 33.3 b | 40.6 b | 27.8 c | 41.2 d | 43.4 e | 52.5 d |
| *R. tropici* | 2.7 c | 5.6 b | 36.7 b | 42.6 b | 29.8 c | 43.4 c | 47.0 c | 54.5 c |
| *R. tropici + A. brasilense* | 3.5 b | 5.7 b | 40.4 a | 56.7 a | 33.7 b | 45.5 b | 49.3 b | 57.0 b |
| *R. tropici + B. subtilis* | 4.6 a | 7.8 a | 42.8 a | 45.5 b | 39.6 a | 48.5 a | 54.5 a | 60.7 a |
| *R. tropici + P. fluorescens* | 3.5 b | 5.5 b | 36.6 b | 43.5 b | 42.0 a | 43.7 c | 48.8 b | 53.7 d |
| *R. tropici + A. brasilense + B. subtilis* | 3.5 b | 4.8 c | 35.2 c | 43.3 b | 39.8 a | 42.8 c | 47.1 c | 55.2 c |
| *R. tropici + A. brasilense + P. fluorescens* | 2.7 c | 4.2 c | 34.5 b | 39.8 b | 30.0 c | 41.6 d | 45.8 d | 53.1 d |
| F-values | | | | | | | | |
| Zn | 86.2 ** | 112 ** | 6.7 * | 9.1 * | 85.4 ** | 239 ** | 5.3 * | 67.3 ** |
| I | 25.3 ** | 23 ** | 9.2 ** | 12.4 ** | 25.9 ** | 28.4 ** | 24.1 ** | 48.1 ** |
| Zn x I | 7.8 ** | 20 ** | 0.9 ns | 0.4 ns | 5.8 * | 1.2 ns | 1.8 ns | 1.6 ns |
| CV (%) | 10.7 | 12.3 | 8.6 | 10.2 | 9.2 | 3.0 | 4.35 | 2.09 |

Means in the column followed by different letters are significantly different (*p*-value ≤ 0.05); ** and *—significant at $p < 0.01$ and $p < 0.05$, respectively; ns—non-significant, by F-test.

The soil Zn application and diazotrophic bacterial co-inoculation and their interactions significantly influenced soil Zn concentration after crop harvest for the study years of 2019 and 2020 (Table 2). The concentration of Zn in the soil after crop harvest was elevated with soil Zn application in combination with diazotrophic bacterial inoculations. Soil Zn application in side dressing had increased soil Zn concentration by 31.1 and 42.2% when compared with no Zn application in 2019 and 2020, respectively. Inoculation of seeds with *R. tropici + B. subtilis* prominently increased soil Zn concentration by 67 and 62% in both years, respectively, in comparison to non-inoculated treatments. The interaction of soil applied Zn and bacterial inoculation were also significant (Figure 3A,B) in both 2019 and 2020. In addition, co-inoculation of *R. tropici + B. subtilis* in 2019 resulted in higher soil Zn concentration, irrespective of Zn fertilization, whereas in 2020, higher soil Zn concentration was observed for *R. tropici + P. fluorescens* in the treatments without Zn application (Figure 3A,B).

The analysis indicated that Zn leaf concentrations were different in both of the study years. The leaf Zn concentration of common bean was significantly influenced by soil Zn application and diazotrophic bacteria co-inoculations, whereas their interactions were insignificant (Table 2). The leaf Zn concentration was higher by 8.6% with soil Zn application in 2020 and 6.2% in 2019, in comparison to plots without Zn being applied. The effect of diazotrophic bacteria co-inoculations was also significant and indicated that Zn leaf concentration was increased by 29 and 40%, respectively, with co-inoculation of *R. tropici + B. subtilis* in both studied years. The leaf Zn concentration was 38% higher in 2020 than in 2019 for the treatments of *R. tropici + B. subtilis*. In both years, the interaction of soil Zn application and bacterial co-inoculations was not significant for leaf Zn concentration.

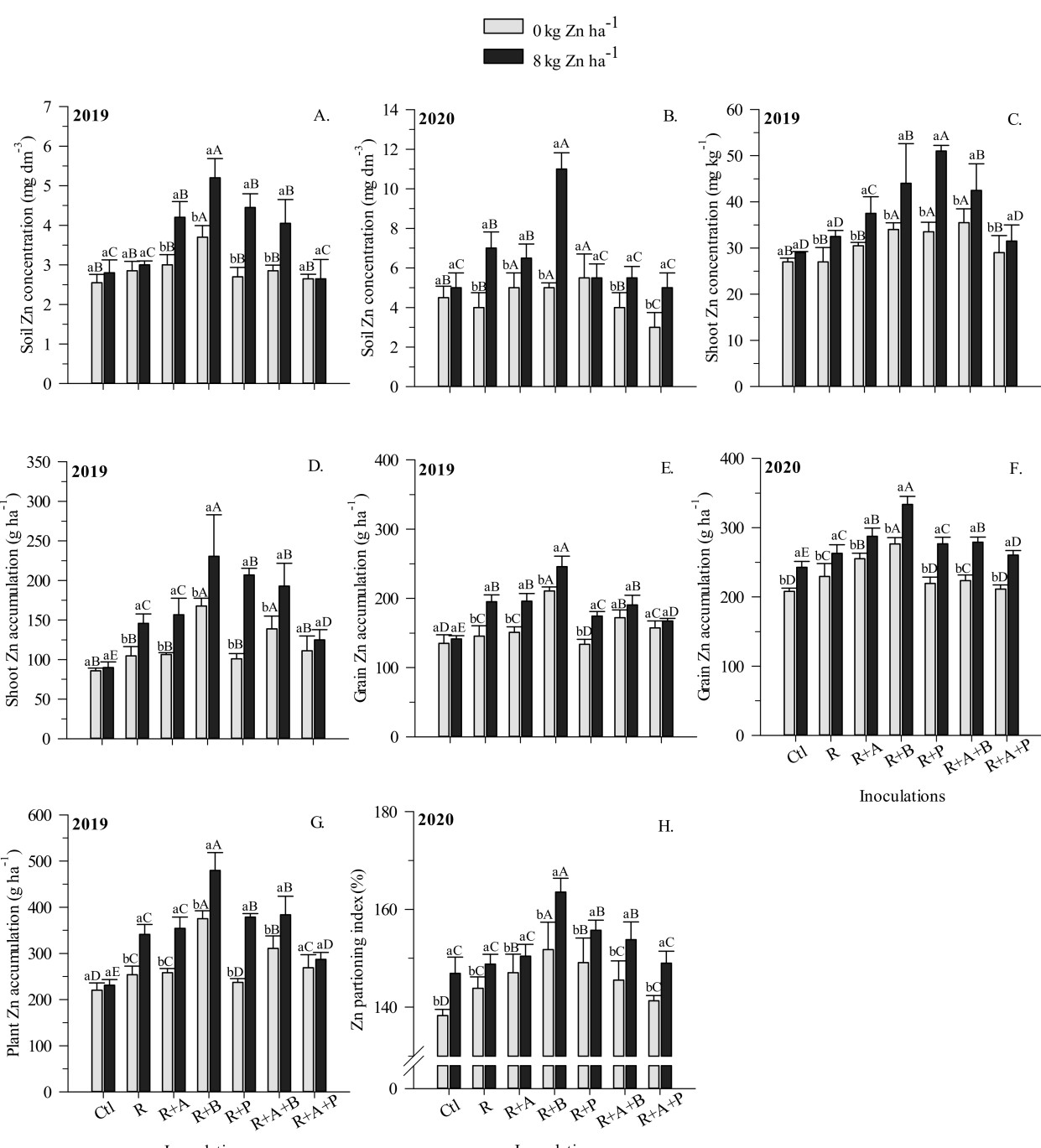

**Figure 3.** The influence of soil Zn and co-inoculations of different diazotrophic bacteria on soil Zn concentration, plant and grain Zn concentration and accumulation, and Zn partitioning index in common bean. (**A**,**B**) Soil Zn concentration (mg dm$^{-3}$) in 2019 and 2020, respectively; (**C**) Shoot Zn concentration (mg) in 2019; (**D**) Shoot Zn accumulation (g kg$^{-1}$) in 2019; (**E**,**F**) Grain Zn accumulation (g kg$^{-1}$) in 2019 and 2020, respectively; (**G**) Plant Zn (straw + grain) accumulation in 2019; and (**H**) Zn partitioning index (%) in 2020. Ctl (No inoculation); R (*R. tropici*); R + A (*R. tropici + A. brasilense*); R + B (*R. tropici + B. subtilis*); R + P (*R. tropici + P. fluorescens*); R + A + B (*R. tropici + A. brasilense + B. subtilis*); and R + A + P (*R. tropici + A. brasilense + P. fluorescens*). The uppercase letters are used for inoculation interactions within each level of soil Zn application, whereas lowercase letters are used for the unfolding of Zn levels within each inoculation treatment. The identical alphabetic letters do not differ from each other, as analyzed by Tukey (Zn application; *p* < 0.05) and Scott–Knott (inoculations; *p* < 0.05) tests for 2019 and 2020, respectively. Error bars indicate the standard error of the mean (n = 4 replications). Selvíria, 2020.

Shoot Zn concentration was improved with soil applied Zn and co-inoculation of diazotrophic bacteria in both of the 2019 and 2020 cropping seasons (Table 2). Application of Zn in side dressing improved shoot Zn concentration by 26 and 13.3% in 2019 and 2020, respectively, when compared to the treatments without Zn application. The co-inoculation of *R. tropici* + *P. fluorescens* improved Zn shoot concentration by 50.6% in 2019, which is statistically similar to values obtained with co-inoculations of *R. tropici* + *A. brasilense* + *B. subtilis* and *R. tropici* + *B. subtilis*. The Zn shoot concentration in 2020 was improved by 17.6% with co-inoculation of *R. tropici* + *B. subtilis*. The interaction of soil applied Zn and bacterial co-inoculations were significant for Zn shoot concentration in 2019 (Figure 3C), whereas the interactions of 2020 were not significant (Table 2). However, triple co-inoculation of *R. tropici* + *A. brasilense* + *B. subtilis* was also observed to have higher shoot Zn concentration in the treatments without soil Zn application (Figure 3C).

Grain Zn concentration of common bean was significantly influenced by soil Zn fertilization and co-inoculations of different diazotrophic bacteria in both 2019 and 2020 (Table 2). The application of Zn in side dressing was noted to have higher grain Zn concentration (increased in by 2.7 and 4.7%, respectively) as compared to untreated plots. The co-inoculation of *R. tropici* + *B. subtilis* was observed to have higher Zn grain concentrations (25.7 and 15.7% for 2019 and 2020, respectively) in comparison to the control treatments. The interactions for grain Zn concentrations in both years were not significant (Table 2).

*3.2. Zinc Accumulation in Plant and Grains, and Partitioning Index (ZPI)*

Zinc accumulation in shoot, plant, and grains and the Zn partitioning index of common beans had a positive relation with soil Zn application and co-inoculation of different diazotrophic bacteria in both cropping seasons (Table 3).

**Table 3.** Zinc accumulation in shoot, grain, and plant (straw + grain), and Zn partitioning index of common bean, as influenced by diazotrophic bacteria and soil Zn application. Selvíria—MS, Brazil, 2019 and 2020.

| Treatments | Zn-Shoot Accumulation | | Zn Grain Accumulation | | Zn-Plant Accumulation | | Zn Partitioning Index | |
|---|---|---|---|---|---|---|---|---|
| | **g ha$^{-1}$** | | | | | | **%** | |
| | **2019** | **2020** | **2019** | **2020** | **2019** | **2020** | **2019** | **2020** |
| Zinc (Zn) application (kg ha$^{-1}$) | | | | | | | | |
| 0 | 116 b | 161 b | 159 b | 233 b | 276 b | 394 b | 122 b | 145 b |
| 8 | 166 a | 189 a | 187 a | 277 a | 340 a | 466 a | 133 a | 152 a |
| Diazotrophic bacterial inoculations (I) | | | | | | | | |
| Without | 89 d | 162 c | 140 e | 223 e | 244 f | 386 d | 126 b | 142 e |
| *R. tropici* | 124 c | 176 b | 173 b | 250 c | 295 d | 426 c | 126 b | 146 d |
| *R. tropici* + *A. brasilense* | 133 c | 191 a | 170 b | 271 b | 300 d | 461 b | 123 b | 149 c |
| *R. tropici* + *B. subtilis* | 202 a | 188 a | 228 a | 307 a | 386 a | 496 a | 137 a | 158 a |
| *R. tropici* + *P. fluorescens* | 153 b | 174 b | 155 d | 246 c | 318 c | 420 c | 125 b | 152 b |
| *R. tropici* + *A. brasilense* + *B. subtilis* | 170 b | 173 b | 180 b | 252 c | 338 b | 425 c | 129 b | 149 c |
| *R. tropici* + *A. brasilense* + *P. fluorescens* | 119 c | 163 c | 162 c | 234 d | 277 e | 397 d | 129 b | 144 d |
| F-values | | | | | | | | |
| Zn | 122 ** | 196 ** | 97.1 ** | 341 ** | 208 ** | 507 ** | 54.3 ** | 138 ** |
| I | 38 ** | 17 ** | 55.6 ** | 75.9 ** | 60.5 ** | 79 ** | 5.0 * | 39 ** |
| Zn x I | 7.3 ** | 1.5 ns | 6.5 * | 3.7 * | 13.6 ** | 1.56 ns | 1.42 ns | 2.7 * |
| CV (%) | 12.01 | 4.24 | 6.16 | 3.50 | 5.35 | 2.78 | 4.38 | 1.57 |

Means in the column followed by different letters are significantly different ($p$-value ≤ 0.05); ** and *—significant at $p < 0.01$ and $p < 0.05$, respectively; and ns—non-significant, by F-test.

The treatment with soil Zn application in side dressing significantly improved shoot Zn accumulation in 2019 and 2020 by 43.1 and 17.4%, respectively, as compared to plots without Zn being applied. The co-inoculation of *R. tropici* + *B. subtilis* increased shoot Zn accumulation by 127% in 2019 whereas, in 2020, shoot Zn accumulation was improved by 18% with co-inoculation of *R. tropici* + *A. brasilense*, which is statistically similar to co-inoculation of *R. tropici* + *B. subtilis*. The interaction of Zn and the co-inoculation of bacteria in 2019 was significant (Figure 3D), although it was not significant in 2020 (Table 3). The co-inoculation of *R. tropici* + *B. subtilis* also lead to greater shoot Zn accumulation in the treatments without soil Zn application (Figure 3D).

Grain Zn accumulation was improved in 2019 and 2020 by 17.6 and 18.9%, respectively, with soil Zn application compared to without soil Zn application. The co-inoculation of *R. tropici* + *B. subtilis* increased grain Zn accumulation by 62.8% in 2019 and 37.7% in 2020, respectively, compared to un-inoculated treatments. The interactions of soil applied Zn and co-inoculation of bacteria were also significant for both the years (Figure 3E,F). The co-inoculation of *R. tropici* + *B. subtilis* with and without soil Zn application was noted for higher grain Zn accumulation, whereas co-inoculation of *R. tropici* + *P. fluorescens* in the absence of Zn application resulted in lower grain Zn concentrations in both 2019 and 2020 (Figure 3E,F).

Plant (straw + grain) Zn accumulation was influenced by soil Zn application and co-inoculation of different diazotrophic bacteria in both of the 2019 and 2020 study years (Table 3). Higher plant Zn accumulations (23.2 and 18.3%) was noted in the plots treated with 8 kg Zn ha$^{-1}$ as compared to the treatments without Zn application in 2019 and 2020, respectively. The co-inoculation of *R. tropici* + *B. subtilis* also increased plant Zn accumulation in 2019 and 2020 (58.2 and 28.5%, respectively) in comparison to there being no inoculation treatments. The interactions of soil Zn application and bacterial co-inoculation were found to be significant for plant Zn accumulation in 2019 (Figure 3G), whereas it was not significant in 2020 (Table 3).

The Zinc partitioning index (ZPI) of the grain was significantly increased with soil Zn application and co-inoculation by different diazotrophic bacteria in 2019 and 2020 (Table 3). The interaction of soil Zn application and co-inoculation of bacteria was not significant in 2019, although, in 2020, it was significant for the ZPI (Figure 3H). The side dressing of Zn at a rate of 8 kg Zn ha$^{-1}$ significantly increased ZPI by 9 and 5% in 2019 and 2020, respectively, compared to there being no Zn applied treatments (Table 3). The co-inoculation of *R. tropici* + *B. subtilis* predominantly boosted ZPI by 8.7 and 17.5% in 2019 and 2020, respectively, when compared to non-inoculated plots. In addition, co-inoculation of *R. tropici* + *B. subtilis* was also observed to have higher ZPI, even in the absence of Zn application, whereas the lower ZPI was observed in the control treatments (Figure 3H).

### 3.3. Plant Height, Dry Matter, Grain Yield and Zn Intake

Plant height of common bean was one of the determining attributes that was significantly influenced by soil Zn application and co-inoculations of different bacteria in 2019, whereas in 2020, plant height was not significantly influenced (Table 4). The plots treated in 2019 and 2020 with Zn fertilizer were seen to have taller plants (7.36 and 5.43%, respectively) compared to those not fertilized with Zn. In 2019, the co-inoculation with *R. tropici* + *B. subtilis* led to taller plants (13%), which was significantly similar to co-inoculation with *R. tropici* + *A. brasilense* and *R. tropici* + *A. brasilense* + *B. subtilis* (12.6%). The plant height in 2020 was statistically not different; however, taller plants were observed in the control and the triple co-inoculation of *R. tropici* + *A. brasilense* + *P. fluorescens*. The interaction for plant height in 2019 was significant (Figure 4A), and that of 2020 was insignificant (Table 4). In addition, treatments in the absence of Zn fertilizer were observed, with taller plants under co-inoculation of *R. tropici* + *A. brasilense* and shorter plants with co-inoculation of *R. tropici* + *P. fluorescens* (Figure 4A).

**Table 4.** Plant height, shoot dry matter, grain yield, and Zn intake in common bean under the influence of diazotrophic bacteria and soil applied zinc doses. Selvíria—MS, Brazil, 2019 and 2020.

| Treatments | Plant Height | | Shoot Dry Matter | | Grain Yield | | Zn Intake | |
|---|---|---|---|---|---|---|---|---|
| | cm | | kg ha$^{-1}$ | | | | g Person$^{-1}$ Day$^{-1}$ | |
| | **2019** | **2020** | **2019** | **2020** | **2019** | **2020** | **2019** | **2020** |
| Zinc (Zn) application (kg ha$^{-1}$) | | | | | | | | |
| 0 | 95 b | 92 a | 3934 b | 3930 b | 3758 b | 4303 b | 6.73 b | 7.68 b |
| 8 | 102 a | 97 a | 4040 a | 4075 a | 4269 a | 4888 a | 6.92 a | 8.04 a |
| Diazotrophic bacterial inoculations (I) | | | | | | | | |
| Without (Control) | 92.1 b | 99.8 a | 3843 d | 3818 d | 3268 e | 4252 e | 6.16 e | 7.46 d |
| *R. tropici* | 98.6 b | 94.1 a | 3952 c | 3989 c | 4121 b | 4590 c | 6.68 c | 7.75 c |
| *R. tropici + A. brasilense* | 103.7 a | 95.4 a | 4060 b | 4063 b | 3899 c | 4748 b | 7.01 b | 8.11 b |
| *R. tropici + B. subtilis* | 104.1 a | 93.0 a | 4207 a | 4164 a | 5069 a | 5052 a | 7.75 a | 8.64 a |
| *R. tropici + P. fluorescens* | 92.2 b | 90.1 a | 3978 c | 4018 b | 3542 d | 4564 c | 6.95 b | 7.64 d |
| *R. tropici + A. brasilense + B. subtilis* | 103.7 a | 94.1 a | 3962 c | 4012 b | 4234 b | 4558 c | 6.70 c | 7.85 c |
| *R. tropici + A. brasilense + P. fluorescens* | 96.0 b | 98.1 a | 3905 d | 3955 c | 3962 c | 4405 d | 6.52 d | 7.55 d |
| F-values | | | | | | | | |
| Zn | 24.1 ** | 3.4 ns | 23.6 ** | 74.9 ** | 161 ** | 306 ** | 5.3 * | 67.2 ** |
| I | 7.2 ** | 0.85 ns | 16.4 ** | 22.7 ** | 115 ** | 33.1 ** | 22.1 * | 48.2 ** |
| Zn x I | 4.8 * | 0.84 ns | 0.13 ns | 2.68 * | 10.2 ** | 5.1 ** | 1.8 ns | 1.7 ns |
| CV (%) | 5.71 | 10.44 | 2.06 | 1.56 | 3.76 | 2.72 | 4.35 | 2.09 |

Means in the column followed by different letters are significantly different ($p$-value $\leq 0.05$); ** and *—significant at $p < 0.01$ and $p < 0.05$, respectively; and ns—non-significant, by F-test.

Soil Zn application at sowing and co-inoculation of different diazotrophic bacteria significantly influenced the shoot dry matter of common bean in both of the 2019 and 2020 study years (Table 4). Shoot dry matter was increased by 2.7 and 3.6% in the presence of Zn fertilization in 2019 and 2020, respectively, compared to the treatments without Zn application. The co-inoculation of *R. tropici + B. subtilis* increased shoot dry matter by 9.5 and 9.1% in 2019 and 2020, respectively, compared to the control. The interaction for shoot dry matter in the first season was not significant (Table 4), whereas, in the second crop season, interaction was significant (Figure 4B).

Grain yield of common bean was significantly influenced by soil Zn application in combination with different diazotrophic bacteria in 2019 and 2020 (Table 4). The interactions of Zn and co-inoculations of bacteria were significant for both years (Figure 4C,D). Grain yield of common bean was increased by 13.6 and 13.5% in 2019 and 2020, respectively, under the treatments with soil applied Zn when compared to the treatments without Zn application. The co-inoculation of *R. tropici + B. subtilis* increased the grain yield of common bean by 54.3 and 18.9% in 2019 and 2020, respectively, compared to the control plots. In addition, co-inoculation of *R. tropici + B. subtilis* was observed for higher grain yield, irrespective of Zn application in both study years (Figure 4C,D). However, lower grain yield was observed with co-inoculation of *R. tropici + P. fluorescens* in 2019 (Figure 4C) and without inoculation in 2020 (Figure 4D).

The estimation of daily Zn intake in Brazil was increased with soil Zn application and co-inoculations of diazotrophic bacteria in 2019 and 2020 (Table 4). The daily consumption of beans in Brazil (~142.2 g person$^{-1}$ day$^{-1}$) [45] and the results of present grain Zn concentration and soil Zn application, along with different diazotrophic co-inoculations, are being calculated to attain Zn intake (g person$^{-1}$ day$^{-1}$). The treatments with soil applied Zn were observed to have higher estimated Zn intake in 2019 and 2020 (2.83 and 4.69%, respectively). The co-inoculation of *R. tropici + B. subtilis* increased estimated Zn intake by 25.8% in 2019 and 15.8% in 2020, compared to the control. The interaction of soil

Zn application and diazotrophic bacteria was not significant for Zn intake in both of the study years.

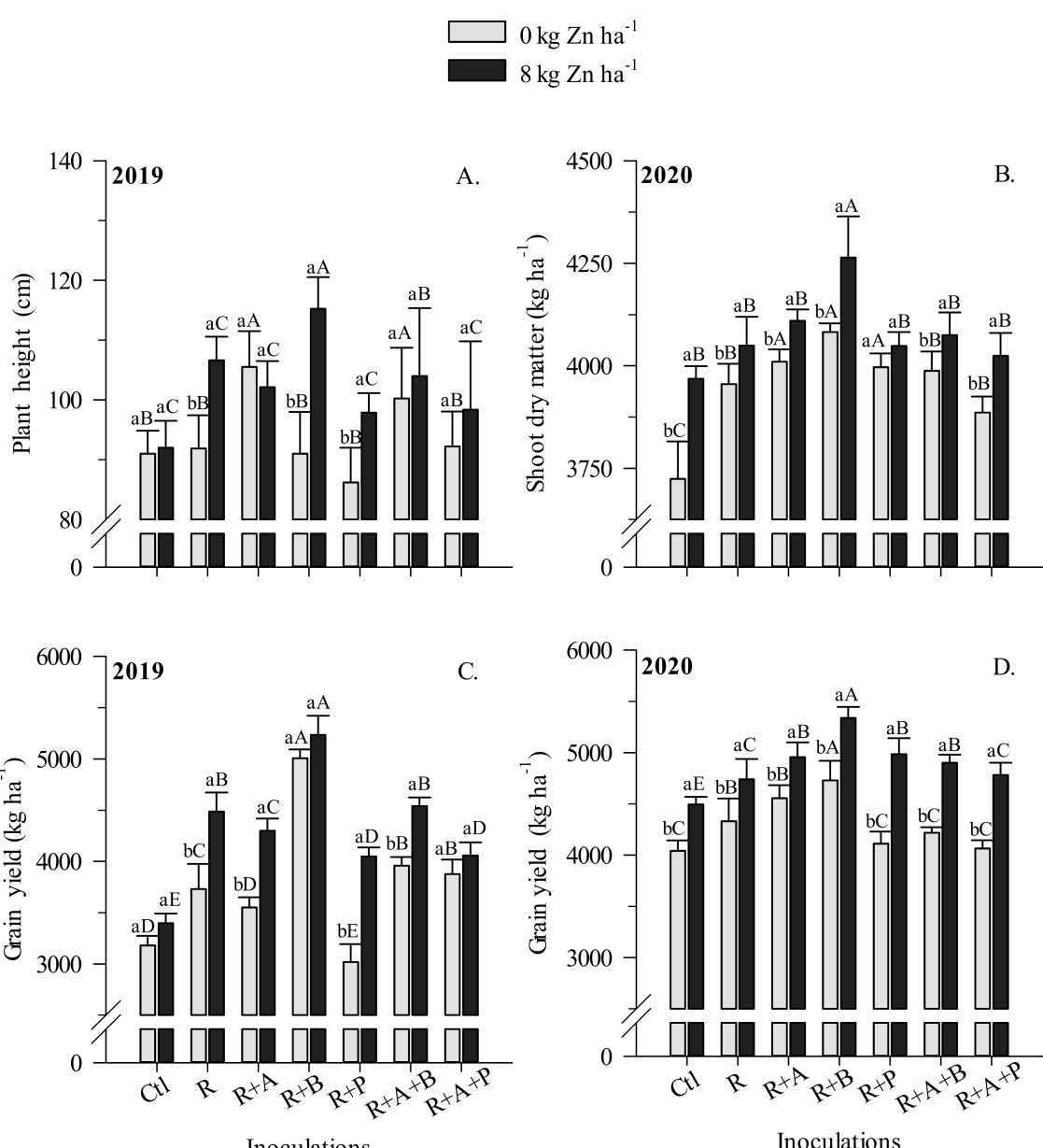

**Figure 4.** The influence of soil Zn and co-inoculations on common bean growth attributes and grain yield: (**A**) Plant height (cm) in 2019; (**B**) Shoot dry matter (kg ha$^{-1}$) in 2020; (**C,D**) Grain yield (kg ha$^{-1}$) in 2019 and 2020, respectively. Ctl (No inoculation); R (*R. tropici*); R + A (*R. tropici + A. brasilense*); R + B (*R. tropici + B. subtilis*); R + P (*R. tropici + P. fluorescens*); R + A + B (*R. tropici + A. brasilense + B. subtilis*); and R + A + P (*R. tropici + A. brasilense + P. fluorescens*). The uppercase letters are used for inoculation interactions within each level of soil Zn application, whereas lowercase letters are used for the unfolding of Zn levels within each inoculation treatment. The identical alphabetic letters do not differ from each other, as analyzed by Tukey (Zn application; $p < 0.05$) and Scott–Knott (inoculations; $p < 0.05$) tests for 2019 and 2020, respectively. Error bars indicate the standard error of the mean (n = 4 replications). Selvíria, 2019 and 2020.

### 3.4. Zinc Efficiencies

Zinc use efficiency (ZnUE) was increased in the treatments of different diazotrophic bacteria in combination with soil Zn application (Table 5). The effect of bacterial co-inoculation was significant for ZnUE in both years. The co-inoculation of *R. tropici* +

*B. subtilis* improved ZnUE in 2019 and 2020 (941.6 and 172%, respectively), compared to control treatments.

**Table 5.** Zinc efficiencies of common beans under the influence of diazotrophic bacteria and soil applied zinc doses. Selvíria—MS, Brazil, 2019 and 2020.

| Treatments | ZnUE | | APE | | UE | | AZnR | |
|---|---|---|---|---|---|---|---|---|
| | kg kg$^{-1}$ | | | | | | % | |
| | **2019** | **2020** | **2019** | **2020** | **2019** | **2020** | **2019** | **2020** |
| Diazotrophic bacterial inoculations (I) | | | | | | | | |
| Without (Control) | 24 d | 64 d | 6.95 a | 5.15 c | 41 d | 133 c | 3 e | 12 d |
| *R. tropici* | 168 b | 104 c | 9.88 a | 6.36 b | 209 b | 183 b | 17 c | 16 c |
| *R. tropici + A. brasilense* | 140 b | 123 b | 7.68 a | 5.90 b | 228 b | 208 b | 18 c | 21 b |
| *R. tropici + B. subtilis* | 250 a | 174 a | 7.17 a | 6.56 b | 447 a | 273 a | 35 a | 26 a |
| *R. tropici + P. fluorescens* | 106 c | 132 b | 4.96 a | 7.66 a | 212 b | 204 b | 21 b | 17 c |
| *R. tropici + A. brasilense + B. subtilis* | 167 b | 121 b | 7.68 a | 7.02 a | 238 b | 198 b | 22 b | 17 c |
| *R. tropici + A. brasilense + P. fluorescens* | 106 c | 104 c | 10.20 a | 7.48 a | 148 c | 175 b | 10 d | 14 d |
| F-values | | | | | | | | |
| I | 51 * | 16 * | 2.96 | 9.10 * | 44.2 * | 13.4 * | 62.2 * | 32.4 * |
| CV (%) | 14.2 | 13.9 | 26.7 | 8.9 | 16.8 | 11.8 | 13.6 | 9.2 |

ZnUE = Zinc use efficiency, APE = Agro-physiological efficiency, UE = Utilization efficiency, and AZnR = Applied zinc recovery. Means in the column followed by different letters are significantly different ($p$-value $\leq$ 0.05); * —significant at $p < 0.01$ and $p < 0.05$, respectively; and ns—nonsignificant, by F-test.

The co-inoculations of different bacteria along with Zn application had not significantly affected agro-physiological efficiency (APE) in 2019, whereas the effect was significant in 2020 (Table 5). The highest APE was observed with the triple co-inoculation of *R. tropici + A. brasilense + P. fluorescens*, which was 46.7% higher than the control, whereas the lowest APE was observed with *R. tropici + P. fluorescens*, which was 0.71% less than the control in 2019. The APE was improved by 48.7% with triple co-inoculation of *R. tropici + A. brasilense + P. fluorescens*, which is statistically similar to co-inoculation of *R. tropici + P. fluorescens* (45.2%) and *R. tropici + A. brasilense + B. Subtilis* (36.3%) in 2020, compared to the control plots.

The utilization efficiency (UE) was significantly improved with co-inoculations of different diazotrophic bacteria along with soil Zn application in 2019 and 2020 (Table 5). The co-inoculation of *A. brasilense + B. subtilis* prominently increased utilization efficiency in 2019 and 2020 by 990.2 and 105.2%, respectively, compared to the control.

The co-inoculations of different diazotrophic bacteria along with soil applied Zn improved applied Zn recovery (AZnR) in 2019 and 2020 (Table 5). The co-inoculation of *R. tropici + A. brasilense* in 2019 and 2020 significantly improved AZnR by 167 and 117%, respectively, along with soil application of Zn, compared to the control.

### 3.5. Pearson's Linear Correlation among Zn Content in Tissues and Soil, Zn Efficiencies and Common Bean Grain Yield

Pearson's linear correlation was positive between Zn content in soil; Zn concentration in leaf tissue, shoot and grain; Zn accumulated in shoot, grain, and aerial part with shoot biomass; and grain yield (Figure 5). Similarly, Pearson's correlation was positive between Zn use efficiency, agro-physiological efficiency, applied Zn recovery, and utilization efficiency with shoot biomass and common bean grain yield (Figure 5).

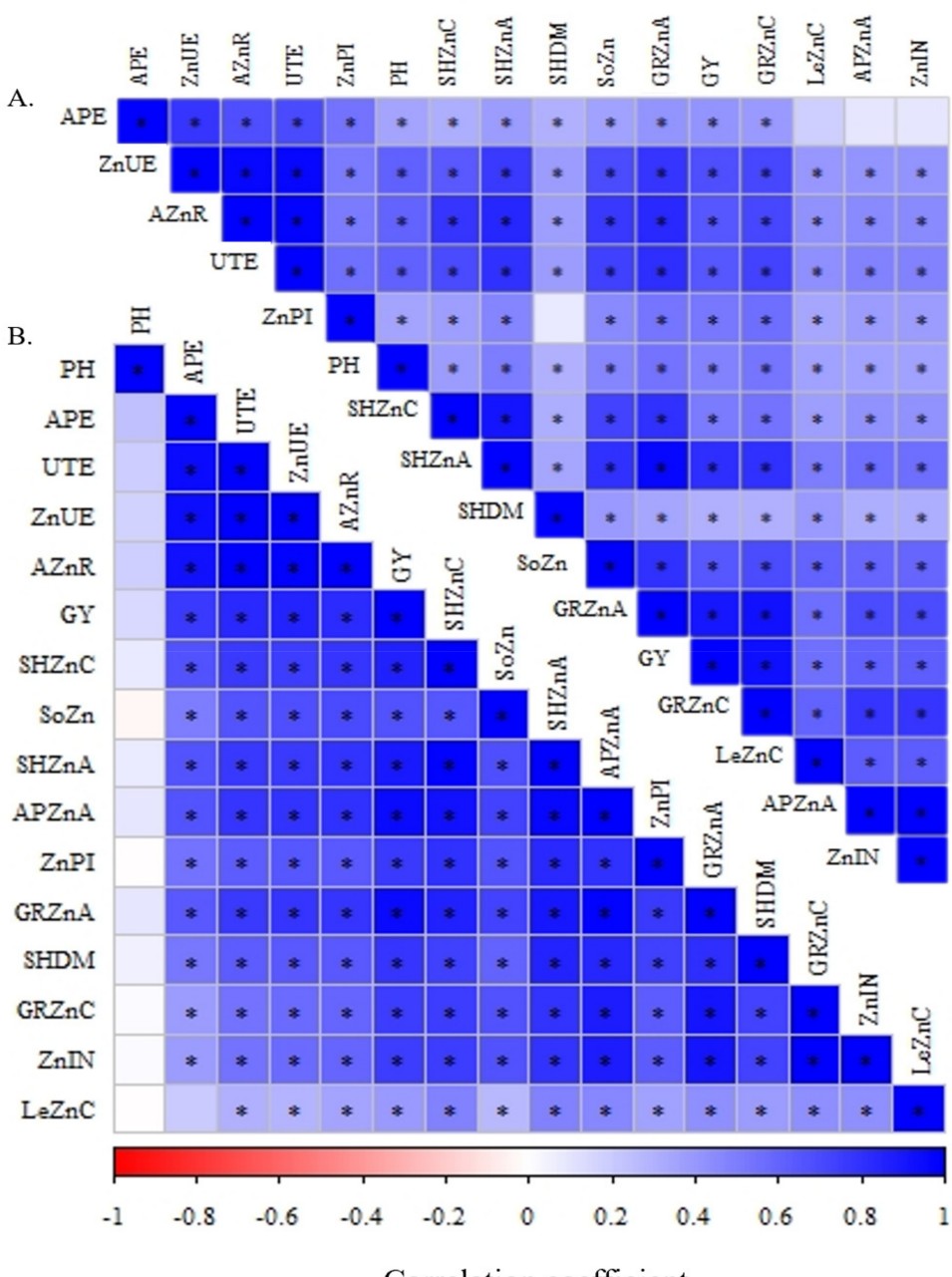

**Figure 5.** Heatmap showing Pearson's correlation among the analyzed parameters from soil and common bean plants in response to different soil Zn applications and diazotrophic bacteria inoculations in the 2019 (**A**) and 2020 (**B**) crop seasons. * indicates a significant relationship ($p \leq 0.05$). Abbreviations: SoZn = Zn content in soil, LeZnC = Zn concentration in leaf tissue, SHZnC = Zn concentration in shoot, SHZnA = Zn accumulated in shoot, APZnA = Zn accumulated in aerial part (shoot + grain), GRZnC = Zn concentration in grain, GRZnA = Zn accumulated in grain, SHDM = shoot dry matter, GY = grain yield, PH = plant height, ZnPI = Zn partioning index, ZnIN = Zn intake, ZnUE = Zn use efficiency, APE = Agro-physiological efficiency, AZnR = Applied Zn recovery, and UTE = Utilization efficiency.

## 4. Discussion

Zinc soil application is one the most adaptable strategies to boost Zn assimilation in plant tissues and grains [26] in Zn deficient soil ranging from 0.0 to 0.3 mg dm$^3$ [46]. In addition, Zn agronomic application improves different physiological functions and results in better growth, Zn use efficiency, and high productivity [20,22,26]. The positive

Pearson's correlation between Zn uptake (Zn concentration in soil and tissues and Zn accumulated in aerial part) and Zn efficiencies with common bean shoot dry matter and grain yield support this hypothesis (Figure 5).

The inoculation of diazotrophic bacteria can improve Zn solubilization and availability to plants through the production of several phytohormones and enzymes and the biological fixation of nitrogen [23,29]. These bacteria facilitate Zn assimilation and accumulation through carboxylation and solubilization, unlike sole Zn applications that generate Zn toxicity [47]. Zinc is available in different ways over time, which needs further and extensive study to understand the co-inoculation and Zn dose effect, which will lead to better nutrition, plant physiology, and yield. The soil Zn application with diazotrophic bacteria sustainably increased plant adaptation in tropical regions, leading to the better Zn nutrition, accumulation, and yield of common bean.

This field study indicated that soil Zn application and co-inoculations of diazotrophic bacteria improved Zn soil concentration after common bean harvest (Figure 3A,B) in 2019 and 2020. This improvement might be due to certain metabolic mechanisms that bacteria could adopt to dissolve unavailable soil Zn through synthesis of reactive products from non-reactive materials [48]. The increase in soil Zn concentration may be due to the positive relationships of different bacteria, including *Bacillus* and *Pseudomonas* sp., which potentially improve soil Zn concentration through carboxylation and different organic acids production [34,49].

The concentration of Zn in shoots and grains were improved with soil applied Zn and co-inoculation of different bacteria (Figure 3C; Table 2). The reason for this might be the structural role of bacteria in different metabolic, enzymatic, and biochemical processes [50], which should proactively increase Zn and other nutrients availability to plants [44]. Plants-microbe interaction in root rhizosphere increased Zn solubilization and availability in the plant tissues of legume crops [51]. Previous studies have shown that the sequencing of bacterial inoculants (especially *Bacillus* sp.) had the potential to better solubilize Zn insoluble complexes, including phosphate, oxides, and carbonates, and make them available for plant uptake and therefore improve the growth attributes of leguminous crops [35,52]. Several previous studies have indicated that different strains of *Bacillus*, *Pseudomonas*, and *Rhizobium* sp. improved Zn solubilization and availability in different legumes and cereal crops [53,54]. The co-inoculation of these microorganisms with synthetic Zn fertilizers increased nutrient concentration and uptake, leading to healthy and quality grains with rich nutrient accumulation in bean cultivars [55]. The indigenous Zn solubilizing bacteria are more effective, with potentially better improvements in different phytohormones than in exogenous ones, and are able to sustainably sustain plant and grain concentrations with higher growth, yield, and soil fertility status [56]. In this way, our study also showed that shoot (Figure 3C) and grain (Table 2) Zn concentrations were improved with co-inoculation of *R. tropici* + *B. subtilis* along with 8 kg ha$^{-1}$ of soil applied Zn. The activities and redistribution of different *Bacillus* strains in rhizosphere Zn pools potentially increased plant Zn availability, growth, and assimilation to the grains of legumes and cereals, allowing them to overcome malnutrition [57].

The accumulation of Zn in the shoots, grains, and plants of common bean (Figure 3D–G) were improved with soil applied Zn and co-inoculation of *R. tropici* + *B. subtilis* bacteria. Both the accumulation and concentration of Zn in plant tissues are directly related to the population, availability, and composition of soil micro-biota in the root rhizosphere [58], which, therefore, with some specific strains, increased Zn availability and uptake [57]. The crops on Zn concentrated soils may uptake a lower concentration of Zn due to its oxides and carbonates [59], which should be potentially increased with root-rhizosphere Zn solubilizing microbes. There have been several previous studies with different crops where *Bacillus* strains, alone or in combination with organic and inorganic fertilizers, increased Zn availability and uptake [60,61], which, therefore, potentially combated Zn scarcity and enhanced the productivity and Zn accumulation in plants [62].

The present study also showed that soil Zn application, along with co-inoculation of *R. tropici* and *B. subtilis*, increased Zn partitioning to common bean grain in two consecutive cropping seasons (Table 3 and Figure 3H). Different strains of *Rhizobium* sp. are one of the PGPBs that potentially convert unavailable mineral nutrients into an available form for plant uptake [63]. A previous study, with a range of bacterial isolates, indicated that *Bacillus* and *Pseudomonas* are the strains with the most potential to solubilize Zn soon after their inoculation [47,64]. Different genera of *Bacillus* are widely known for Zn solubilization and mobilization from soil to plant and, subsequently, to grains [34]. Therefore, our study also confirmed that Zn partitioning to the grains was improved with co-inoculation of *R. tropici* and *B. subtilis*. A study described that the Zn partitioning index was regulated up and down with inoculation of *Bacillus* in legume and cereal crops due to the up and down regulation of Zn transporter genes and, therefore, *Bacillus* is one of the most promising candidate bacteria for the bioavailability and bio-fortification in several crop [62,65].

The plant height, dry matter, and grain yield (Figure 4A–D) of common bean were also increased with soil Zn application and co-inoculation of diazotrophic bacteria. Zinc is one of the most imperative nutrient of its kind that affects crop growth and development [66], is known for having indispensable functions in cell division, and contains ribosomal stabilization carbohydrate and several other growth promoting enzymes. The application of Zn favored plant growth by favoring cell division in a better way, Zn being an imperative component of several plant biochemical processes [50]. Zinc has a proactive role in the maintenance of cell integrity, elongation, and multiplication [21], along with enzyme activities [22] that benefit plant physiology and biomass production with high yielding potential [23,67]. The present study supports these previous studies, demonstrating that, in common bean crops, the application of Zn resulted in taller plants with higher dry matter production and yield.

The inoculation of diazotrophic bacteria, especially Zn solubilizing bacteria, play an outstanding role in maintaining a sustainable and eco-friendly environment [28]. This micro-biota biodiversity adopts several mechanisms to not only regulate the environmental cause but also improve plant growth, physiology, and yield, with a better accumulation of nutrients in grains for human benefits [48,64,68]. The same strategy has been followed in our study, with soil Zn application and *R. tropici* and *B. subtilis*, for the increment of shoot dry matter, grain yield (Figure 4B–D) of common bean, and estimated Zn intake. It has been reported that *Bacillus* strains potentially decrease phytic-P concentration in legume grains, which improves Zn bioavailability in seeds for assimilation and consumption by humans, with a high feed efficiency [69]. It should also be considered that Zn rates tested for Zn concentration and intake in the present study are not of concern in term of toxicity for plants, which are recognized as being safe for common bean biofortification. This is also be due to the bio-activation of several zinc-soil-microbes-plant mechanisms to solubilize unavailable soil Zn and improve plant Zn uptake and biofortification [53,57,70]. Therefore, our results also highlight that soil Zn application has synergistically improved the nutritional status and yield of common bean when co-inoculated with *R. tropici* and *B. subtilis* (Table 4; Figure 4C,D).

The Zn efficiencies were improved with soil Zn application and bacterial co-inoculation (Table 5). Zinc efficiency is determined in term of Zn availability in the grains of low Zn available soils [40,44]. Studies determining the impact of applying Zn from 0 to 10 kg ha$^{-1}$ have demonstrated the potentially improved nutritional quality of bean grains, without toxicity symptoms, in tropical soil [71]. The sole application of Zn may lead to a negative relation with Zn use efficiencies [72]; however, inoculant strains of *Bacillus* sp. and *Pseudomonas* sp. are considered to have high dissolution properties regarding Zn sulphide, oxides, and carbonates [53]. These rhizosphere micro-biota adopt several mechanisms to promote the plant root system, which therefore increase nutrient uptake and utilization. Hence, our results have exhibited that *R. tropici* and *B. subtilis* improved ZnUE, AZnR, and Zn utilization (Table 5). The better Zn use efficiencies with Zn application might be due to better crop establishment and grain yield (Table 4), which was further increased with

Zn assimilation to grains under co-inoculation of these diazotrophic bacteria (Figure 3H). Our study also indicated that agro-physiological were improved with triple co-inoculation (*R. tropici* + *A. brasilense* + *P. fluorescens*) along with soil Zn application (Table 5). This positive improvement in Zn efficiencies with Zn application may be in response to the low available concentration of Zn in the tropical savannah [44]. Therefore, inoculation of Zn solubilizing bacteria increased Zn partitioning and accumulation irrespective of Zn use efficiency and proved a sustainable and integrated strategy for grain biofortification and productivity of the common bean, the most important leguminous crop, which could improve human nutrition in a sustainable way.

## 5. Conclusions

Soil zinc application is one feasible strategy to enrich and fortify the grains of crops. The application of diazotrophic bacteria, including Zn solubilizing bacteria, is an interesting alternative sustainable strategy to encourage Zn use efficiencies in an eco-friendly way. Our results indicated that soil Zn application with different diazotrophic bacteria improved Zn concentrations in the soil, as well as in the plants and grains of common bean. It was also concluded that soil Zn application in combination with co-inoculation of *R. tropici* and *A. brasilense* to common bean sustainably improved Zn-leaf. The co-inoculation of *R. tropici* and *B. subtilis* in combination with Zn better improved Zn concentration and accumulation in shoots and grains, with a promising effect on grain yield and the estimated Zn intake of the common bean. The Zn use efficiencies were prominently improved with co-inoculation, irrespective of Zn application. The Zn use efficiency, applied Zn recovery, and Zn utilization efficiency were higher with co-inoculation of *R. tropici* and *B. subtilis* in comparison to all other inoculations. Therefore, co-inoculation of *R. tropici* and *B. subtilis* could be the most effective method, in association with soil Zn application, for improvement in the acquisition of plant nutrients and their use efficiencies, especially Zn, for the better biofortification of common bean grains in tropical regions. Further studies aimed at improving Zn utilization and recovery in combination with diazotrophic bacteria and their impact on legumes biofortification, sequencing, and physiological processes should be performed under different environmental and edaphic conditions to better understand Zn solubilizing bacteria under field conditions.

**Author Contributions:** A.J. and M.C.M.T.F., conceptualized the project, investigated, collected, and analyzed the original draft of data; M.C.M.T.F., project administration and supervision; E.H.M.B., C.E.d.S.O., and F.S.G., graph editing; A.R.d.R., F.S.G., and T.A.R.N., review and editing; M.J.M.N., E.S.M., and G.C.F., field and lab help. All authors have read and agreed to the published version of the manuscript.

**Funding:** The authors thank the Conselho Nacional de Desenvolvimento Científico e Tecnológico (CNPq) for the productivity grant in research (award number 312359/2017-9), The World Academy of Sciences (TWAS) for the first author's scholarship (CNPq/TWAS grant number: 166331/2018-0), and the Coordenação de Aperfeiçoamento de Pessoal de Nível Superior (CAPES/AUXPE award number 88881.593505/2020-01).

**Institutional Review Board Statement:** Not applicable.

**Informed Consent Statement:** Not applicable.

**Data Availability Statement:** The data presented in this study are available on request from the corresponding author. The data are not publicly available due to the authors' option.

**Conflicts of Interest:** The authors declare no conflict of interest.

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
