# Peer review of "Common Bean Yield and Zinc Use Efficiency in Association with Diazotrophic Bacteria Co-Inoculations"

_agronomy, doi:10.3390/agronomy11050959_

Round 1

Reviewer 1 Report

In this manuscript, the authors have examined the role of diazotrophic bacterial co-inoculations in association with soil Zn application on Zn nutrition, growth, yield and Zn use efficiencies in common bean in 2019 and 2020 crop seasons. This study provides some new insights into the role of co-inoculation of R. tropici + B. subtilis in association with Zn for biofortification and increase the Zn use efficiencies in common bean in tropical savannah of Brazil.

The following points needs to be addressed:

Table 2 - Letters of significance are missing in the data concerning the Zn-soil (2019 and 2020)and Zn-shoot (2019) concentration, in consideration of the described significance.

Figure 3 - Indicate how the error bars are represented (eg standard error of the mean, standard deviation, etc ...).

Table 3 - Letters of significance are missing in the data concerning the Zn-shoot (2019), Zn-grain (2019 and 2020), Zn-plant (2019) accumulation and Zn-partitioning index (2020), in consideration of the described significance.

Table 4 - Letters of significance are missing in the data concerning the Plant heigh (2019), Shoot dry matter (2020) and Grain yield (2019 and 2020), in consideration of the described significance.

Figure 4 - Indicate how the error bars are represented (eg standard error of the mean, standard deviation, etc ...).

Line 245: I recommend to change in-significant to insignificant

Line 251: I recommend to change non-significant to not significant

Line 334: I recommend to change non-significant to not significant

Line 356: I recommend to change in-significant to insignificant

Line 362: I recommend to change nonsignificant to not significant

Line 435: I recommend to change nonsignificant to not significant

Line 493: I recommend to change phyto-hormones to phytohormones

Paragraph 2.6. Statistical analysis. In addition to the Shapiro-Wilk normality test, for a correct use of ANOVA it would also be necessary to perform the homogeneity of variance test (eg. Levene’s test or others). A more comprehensive description of the statistical analysis is necessary. You have an experiment where the factors are Zinc (Zn) application (kg ha-1) and Diazotrophic bacterial inoculations (I) and you collect data in 2019 and 2020. I think that to analyze the data you should use the Split-Plot in time.

Author Response

Dear reviewer, 

We appreciate and thankful to you for taking time from your busy schedule and review our manuscript. 

Our responses to your comments are attached in PDF.  

Reviewer 2 Report

The manuscript entitled "Common bean yield and zinc use efficiency in association with diazotrophic bacteria co-inoculations" is written at a good level, however I have a few suggestions and recommendations:

There are few inconsistencies in the manuscript, which are marked in the text (see attachment).

Besides that:

In the context of the results, an important question arises. Was the common bean grown in monoculture or was it included in the crop rotation? Because the data on the analysis of soil samples from 2020 are missing in section 2.2 (Table 1). It is also interesting that in 2020 the values ​​of all parameters (tables 2 and 3) were significantly higher than in 2019. How do you explain these big differences? If it was grown in monoculture, did it not have a significant effect on the content of Zn in the soil in 2020, and subsequently on all other parameters associated with it? This could then be the subject of discussion.

In some cases, Tables 3 and 4 lack the indication of statistical differences, which are commented on in the text. It is essential that all differences are marked.

It is a pity that in Figures 3 and 4 there are no graphs for the same parameters in both experimental years. Can you explain?

Author Response

(The authors gave the same response as above.)
